# Direction of Arrival Estimation with Nested Arrays in Presence of Impulsive Noise: A Correlation Entropy-Based Infinite Norm Strategy

**Jun Zhao [1], Renzhou Gui [1], Xudong Dong [2],*, Meng Sun [2,3] and Yide Wang [4]**

[1] College of Electronic and Information Engineering, Tongji University, Shanghai 201800, China
[2] College of Electronic and Information Engineering, Nanjing University of Aeronautics and Astronautics, Nanjing 210000, China
[3] Nanjing Panda Electronics Company Limited, Nanjing 210000, China
[4] IETR, University of Nantes, 44306 Nantes, France
* Correspondence: nanhangdxd@nuaa.edu.cn

**Abstract:** Direction of arrival (DOA) estimation with nested arrays has been widely investigated in the field of array signal processing, but most studies assume that the noise is Gaussian white noise. In practical situations, there may exist impulsive noise (a kind of heavy-tailed noise), wherein the performance of traditional subspace-based DOA estimation algorithms deteriorates significantly. In this paper, we propose a correlation entropy-based infinite norm preprocessing algorithm, which can be applicable to any type of impulsive noise. Each snapshot of the sensor array data is processed by an exponential kernel function with the infinite norm, which can effectively combat the outliers. Furthermore, we construct the equivalent second-order covariance matrix and perform DOA estimation using classical subspace methods. Simulation results demonstrate the effectiveness of the proposed method for both symmetric $\alpha$-stable distribution and the Gaussian mixture model.

**Keywords:** direction of arrival (DOA) estimation; impulsive noise; exponential kernel function

## 1. Introduction

Direction of arrival (DOA) estimation has been a research hotspot in array signal processing, which is widely used in wireless communication, sonar, radar, and electronic reconnaissance [1,2]. Some classical DOA estimation algorithms based on subspace classes include multiple signal classification (MUSIC) [3], thereby estimating the signal parameter via rotational invariance techniques (ESPRIT) [4], maximum likelihood (ML) [5] algorithms, etc. In addition, compressive sensing-based methods [6,7] and sparsity-based DOA estimation algorithms [8–11] have also been proposed to solve the problem of grid mismatch. However, all of the abovementioned traditional methods investigate DOA estimation under the assumption of additive Gaussian noise. In practical engineering applications, noise, such as astronomical noise generated in the atmosphere, sparking noise generated by vehicles, electrical and industrial equipment operation noise, etc., will exhibit impulsive and time domain-sparse characteristics [12]. This type of noise is called impulsive noise, which has neither second-order statistics nor higher-order cumulants and cannot be directly processed by the traditional second-order statistics-based methods.

To suppress the outliers of the impulsive noise, the robust covariation-based MUSIC (ROC-MUSIC, [13]) DOA estimation method exploits the covariation matrix of the array sensor outputs and assumes that the signal and additive noise obey a joint symmetric $\alpha$-stable (S$\alpha$S, [14]) distribution. In practice, ROC-MUSIC can also be considered as a method based on fractional-order low-order moments (FLOM, [15]). However, it is unrealistic for the signal and the additive noise to jointly obey the S$\alpha$S distribution, because a practical signal always has finite variance, while the signal that obeys the S$\alpha$S distribution has infinite

variance. In [16]. The sign covariance matrix MUSIC (SCM-MUSIC) method can obtain a convergent estimation of the signal and noise subspaces, which solves the problem that the FLOM-like methods are only applicable to $1 < \alpha < 2$. A new subspace algorithm based on phased fractional low-order moments (PFLOM) was proposed in [17], which can obtain better performance for $0 < \alpha < 1$. The infinity norm normalization MUSIC (IN-MUSIC, [18]) and zero-memory nonlinear (ZMNL, [19]) methods have been proposed to limit the influence of impulsive noise by pruning the amplitude of the received array signal. The IN-MUSIC method provides a more accurate DOA estimation than the FLOM, PFLOM, and SCM methods. However, its performance may deteriorate as the signal subspace rank increases. Recently, based on the correlation entropy property, a series of methods have been proposed to combat impulsive noise, such as correntropy-based correlation (CRCO, [20]), generalized autocorrentropy (GCO, [21]), and operator and generalized maximum complex correntropy criterion (GMCCC, [22]). In addition, sparse representation methods [23], as well as sparse Bayesian learning (SBL, [24,25]) methods have also been developed for DOA estimation in the presence of impulsive noise. Nevertheless, only traditional uniform linear arrays (ULAs) are considered in the above mentioned methods, thus resulting in a limited estimation accuracy.

Sparse arrays have attracted much attention by increasing the degrees of freedom (DOFs) for DOA estimation [26–29], such as the super nested array (SNA, [26,27]), augmented nested array (ANA, [28]) and dilated nested array (DNA, [29]). In [30], a sparse array DOA estimation algorithm based on structured correlation reconstruction was proposed, which implements Nyquist space filling on the physical array and performs a compressive transformation on the equivalent filled array to ensure its general applicability and estimation accuracy. In [31], the authors proposed a coarray tensor DOA estimation algorithm for sparse arrays with multidimensional structures. Nevertheless, the above-mentioned methods do not consider the case where the received signal is interfered by the impulsive noise. Based on this, the sparse array technique [32–35] has also been extended to the impulsive noise scenario [36,37]. In these methods [36,37], the original signal covariance matrix is replaced by the PFLOM matrix in combination with the sparse array technique. Alternatively, we found that it is feasible to replace the PFLOM matrix with fractional-order low-order statistics (FLOSs), such as ROC, FLOM, PFLOM. However, the FLOSs-based methods require a priori knowledge of the impulsive characteristic exponent $\alpha$ and involve the choice of unknown parameters (e.g., the order moment parameter $p$), and the uncertainty of the order moment parameter causes instability in the algorithm performance.

In this paper, we propose a correlation entropy-based infinite norm (Co-IN) strategy to combat the impulsive noise outliers. This strategy belongs to the data-adaptive zero-memory (DA-ZM) algorithmic class and has the following characteristics: (1) It is "blind" and does not require any a priori knowledge of "heavy-tailed" noise statistics. (2) It is applicable to any impulsive noise model, including $S\alpha S$ distributions and Gaussian mixture models. (3) It can estimate parameters from impulsive noise-disturbed data by utilizing a second-order statistics-based algorithm. (4) It is data-adaptive and zero-memory, and it does not require prior information retention or linkage to other snapshots. The main contributions are as follows:

- We propose a data-adaptive zero-memory exponential infinity norm strategy based on correlation entropy to suppress the impulsive noise outliers without the prior information of impulsive noise.
- We analyze the pseudocovariance matrix of the processed signal data and prove its boundedness.
- We extend the proposed Co-IN method to the nested array scenario [34].

## 2. Signal Models with a Nested Array Structure

Consider $Q$ far-field independent narrowband signals impinging on a nested array (shown in Figure 1) with DOA $\theta_q, q = 1, 2, \cdots, Q$. The nested array is composed of two

uniform linear arrays (ULAs) with element spacings of $l_1$ and $l_2 = (N_1 + 1)l_1$, respectively. Assuming that the first array element of the first subarray is the reference point, the locations of the array sensors are then given by

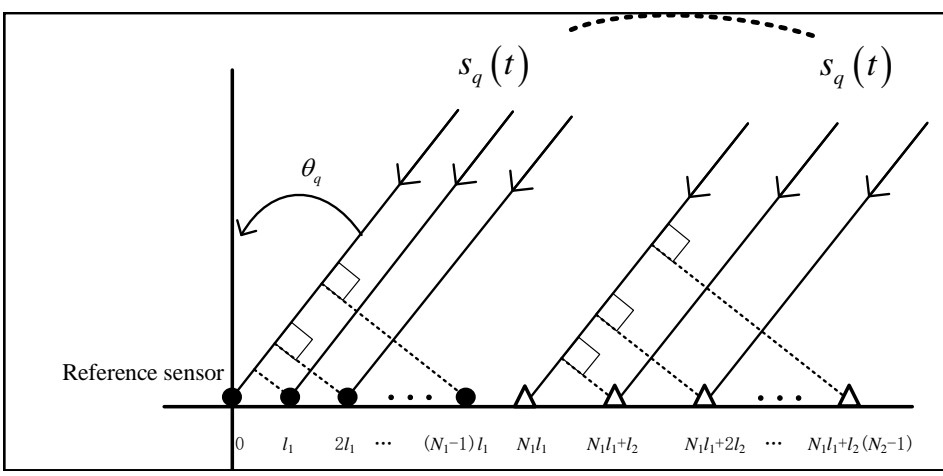

**Figure 1.** Nested array model.

$$
\begin{aligned}
\mathbb{L} &= \{nl_1 | 0 \leqslant n \leqslant N_1 - 1\} \cup \{N_1 l_1 + m l_2 | 0 \leqslant m \leqslant N_2 - 1\} \\
&= \{L_p | p = 1, 2, \cdots, N_1 + N_2, L_1 = 0\},
\end{aligned}
\tag{1}
$$

where $N_1, N_2 \in \mathbb{N}$, $l_1 = \lambda/2$, and $\lambda$ are the carrier wavelengths of incoming signals. The array output ($t = 1, 2, \cdots, T_s$ with $T_s$ being the number of snapshots) is

$$
\mathbf{z}(t) = \sum_{q=1}^{Q} \mathbf{a}(\theta_q) s_q(t) + \mathbf{n}(t) = \mathbf{A}\mathbf{s}(t) + \mathbf{n}(t),
\tag{2}
$$

where $\mathbf{z}(t) = [z_1(t), z_2(t), \cdots, z_{N_1+N_2}(t)]^T$ is the received data vector with the transpose operation $(\cdot)^T$; $\mathbf{A} = [\mathbf{a}(\theta_1), \cdots, \mathbf{a}(\theta_q), \cdots, \mathbf{a}(\theta_Q)]$ is the directional matrix with the steering vector $\mathbf{a}(\theta_q) = [1, e^{-j2\pi L_2 \sin \theta_q / \lambda}, \cdots, e^{-j2\pi L_{N_1+N_2} \sin \theta_q / \lambda}]^T$; $\mathbf{s}(t) = [s_1(t), \cdots, s_Q(t)]^T$ is the signal source vector, and $\mathbf{n}(t) = [n_1(t), n_2(t), \cdots, n_{N_1+N_2}(t)]^T$ is the impulsive noise term obeying $\alpha$-stable distribution [12] or the Gaussian mixture model (GMM) [19].

Since impulsive noise does not have second-order statistics, subspace-based DOA estimation methods, such as MUSIC [3] and ESPRIT [4], will fail in the scenario of impulsive noise. Therefore, it is essential to develop a method for recovering the virtual equivalent second-order statistics that are applicable to any impulsive noise scenario.

### 3. Proposed Method

#### 3.1. Correlation Entropy-Based Infinite Norm (Co-IN) Strategy

Motivated by the information theory, the correlation entropy of two random vectors $\mathbf{x}$ and $\mathbf{y}$ can be defined as

$$
\mathbf{c}_\sigma(\mathbf{x}, \mathbf{y}) = \mathbb{E}\{\kappa_\sigma(\mathbf{x} - \mathbf{y})\},
\tag{3}
$$

where $\kappa_\sigma(\cdot)$ is a kernal function, such as an exponential kernel, a Gaussian kernel, or a Laplacian kernel [21], and $\mathbb{E}\{\cdot\}$ is the mathematical expectation operation. The correlation entropy can convey information about the correlations and statistical distributions of random variables.

Since the received data contain outliers, we propose an autocorrelation entropy (ACE) operator based on the following exponential kernel function

$$
\kappa_\sigma(x) = e^{-\frac{|x|}{2\sigma^2}},
\tag{4}
$$

where $x$ is a scalar, $\sigma$ is the size of kernal, and $|\cdot|$ is the modulus operator. The ACE operator for the received array data in Equation (2) can be expressed as

$$\kappa_\sigma(\mathbf{z}(t)) = \left[ e^{-\frac{|z_1(t)|}{2\sigma^2}}, e^{-\frac{|z_2(t)|}{2\sigma^2}}, \cdots, e^{-\frac{|z_{N_1+N_2}(t)|}{2\sigma^2}} \right]^T. \tag{5}$$

For a finite number of data samples, we have

$$\mathbf{c}_\sigma(\mathbf{Z}) = \mathbb{E}\{\kappa_\sigma(\mathbf{z}(t))\} =$$
$$\frac{1}{T_s} \left[ \sum_{t=1}^{T_s} e^{-\frac{|z_1(t)|}{2\sigma^2}}, \sum_{t=1}^{T_s} e^{-\frac{|z_2(t)|}{2\sigma^2}}, \cdots, \sum_{t=1}^{T_s} e^{-\frac{|z_{N_1+N_2}(t)|}{2\sigma^2}} \right]^T, \tag{6}$$

where $\mathbf{c}_\sigma(\mathbf{Z}) \in \mathbb{C}^{(N_1+N_2)\times 1}$ is a vector.

Subsequently, we consider that the noise follows the symmetric $\alpha$-stable (S$\alpha$S) distribution or the GMM. A histogram with 1000 samples of these two noise distributions is shown in Figure 2a and Figure 2b, respectively. For the S$\alpha$S noise simulation, $\alpha = 1.2$, and the dispersion parameter $\gamma = 0.1$. In the GMM, it consists of two Gaussian noises with $\sigma_2^2 = 100\sigma_1^2$, and the weighting coefficients of the two terms are $c_1 = 0.9$ and $c_2 = 0.1$, respectively. Figure 2a,b show that the amplitudes of a large number of samples are around zero in all the sample data for both noises, while significant outliers are present in only a limited number of samples. In this paper, we propose a correlation entropy-based infinite norm (Co-IN) strategy that does not require any prior information about the noise, which utilizes an exponential infinity norm to compress the outliers of impulsive noise.

**Definition 1.** *Assuming that $\mathbf{z}(t)$ is the received array data with impulsive noise, and $\mathbf{z}(t)$ can be normalized as*

$$\mathbf{y}(t) = w(t)\mathbf{z}(t), \tag{7}$$

*with*

$$w(t) = e^{-\left\| \frac{|\mathbf{z}(t)|}{2\sigma_t^2} \right\|_\infty}$$
$$= e^{-\max\left\{ \frac{|z_1(t)|}{2\sigma_t^2}, \frac{|z_2(t)|}{2\sigma_t^2}, \cdots, \frac{|z_{N_1+N_2}(t)|}{2\sigma_t^2} \right\}}, \tag{8}$$

*and the adaptive kernel size function $\sigma_t$ satisfies*

$$\sigma_t = \frac{2\pi}{3(1+e^{-H_t})}, t = 1, 2, \cdots, T_s, \tag{9}$$

*where $H_t$ is the local entropy for the $t$-th snapshot, which is defined as*

$$H_t = -\sum_{i=1}^{N_1+N_2} f_{it} \log(f_{it}), \tag{10}$$

$$f_{it} = \frac{|z_i(t)|}{\sum_{k=1}^{N_1+N_2} |z_k(t)|}. \tag{11}$$

*Then, the pseudocovariance matrix can be constructed from the $T_s$ samples as follows*

$$\mathbf{R}_{Co\text{-}IN} = \frac{1}{T_s} \sum_{t=1}^{T_s} \mathbf{y}(t)\mathbf{y}^H(t), \tag{12}$$

*and $\mathbf{R}_{Co\text{-}IN}$ is bounded; the proof is provided in Appendix A.*

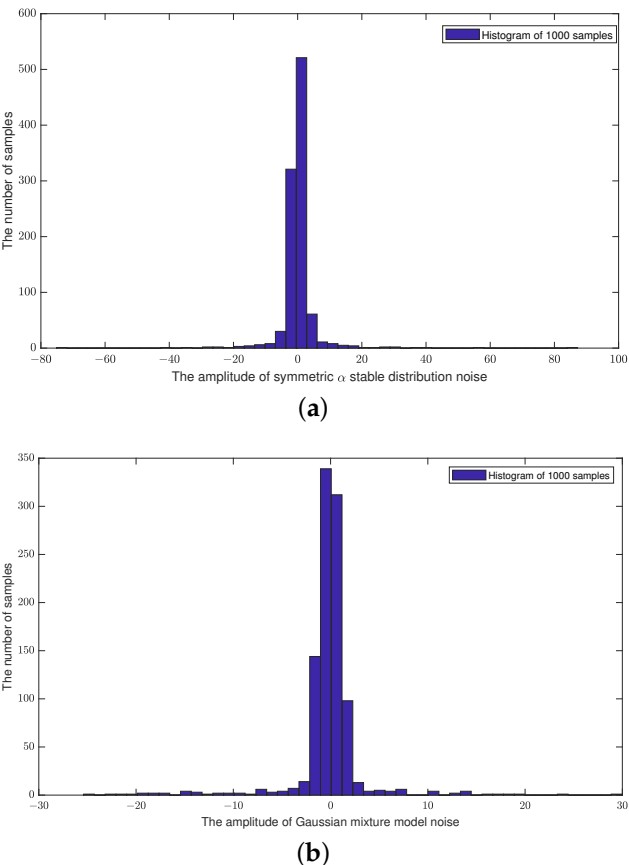

**Figure 2.** Histogram of 1000 samples for S$\alpha$S noise and GMM noise (the horizontal coordinate indicates the amplitude of the received signal samples, and the vertical coordinate represents the number of samples). (**a**) Histogram of S$\alpha$S noise. (**b**) Histogram of GMM noise.

*3.2. Statistical Analysis of the Co-IN Strategy*

The proposed Co-IN strategy can guarantee the zero spatial crosscorrelation of the input noise, provided that the prenormalized impulsive noise follows a spherically symmetrical distribution (This section demonstrates that the proposed Co-IN strategy normalization method preserves the zero spatial crosscorrelation of input noise when the prenormalized noise exhibits a spherically symmetrical distribution on the complex plane, along with defined mean and variance parameters. Examples of such impulse noises are complex value S$\alpha$S noise, as well as special cases of complex-value Gaussian noises and complex-value Gaussian mixture noises, which have many applications in signal processing and communication, such as the noise in telephone circuits, anthropogenic and natural impulsive noise, underwater acoustic signals, and low-frequency atmospheric noise. For details, we can refer to reference [12]) on the complex plane, as has been discussed in [18]. Thus, the prenormalized impulsive noise has no pseudocorrelation between the noise subspace and the signal subspace.

**Corollary 1.** *Considering an independent and S$\alpha$S $(N_1 + N_2) \times 1$ impulsive noise $\mathbf{n}(t)$, there is no signal; then, $\mathbf{z}(t) = \mathbf{n}(t)$, i.e., $\mathbf{y}(t) = w(t)\mathbf{n}(t)$. Then, the proposed Co-IN strategy would give*

$$\mathbb{E}\{\mathbf{y}(t)\} = \mathbf{0}, \tag{13}$$

**0** *means an all-zeros vector, and*

$$\mathbb{E}\left\{\mathbf{y}(t)\mathbf{y}^H(t)\right\} = \mathbf{D}, \tag{14}$$

*where* **D** *is a diagonal matrix with finite entries.*

**Proof.** Because $w(t)$ is real-valued, $w(t)n_i(t)$ is spherically symmetrical, so $\mathbb{E}\{w(t)n_i(t)\} = 0$, $\forall i = 1, \cdots, N_1 + N_2$, and Equation (13) holds.

In the autocorrelation of the noise component, then

$$
\begin{aligned}
&\mathbb{E}\{y_i(t)y_i^*(t)\} \\
&= \mathbb{E}\{w(t)n_i(t)w^*(t)n_i^*(t)\} \\
&= \mathbb{E}\left\{w^2(t)|n_i(t)|^2\right\} \\
&\leqslant \mathbb{E}\left\{e^{-\frac{|n_i(t)|}{\sigma_t^2}} \cdot \frac{|n_i(t)|^2}{\sigma_t^4} \cdot \sigma_t^4\right\},
\end{aligned}
\tag{15}
$$

where $(\cdot)^*$ is the conjugate operation, and $e^{-\frac{|n_i(t)|}{\sigma_t^2}} \times \frac{|n_i(t)|^2}{\sigma_t^4} \times \sigma_t^4 \leqslant \frac{4\sigma_t^4}{e^2}$; thus, $\mathbb{E}\left\{w^2(t)|n_i(t)|^2\right\}$ has a finite value.

Similarly, for $\forall i, j \in 1, 2, \cdots, N_1 + N_2$ and $i \neq j$, we have

$$
\begin{aligned}
&\mathbb{E}\left\{y_i(t)y_j^*(t)\right\} \\
&= \mathbb{E}\left\{w(t)n_i(t)\left(w(t)n_j(t)\right)^*\right\} \\
&= \mathbb{E}\{w(t)n_i(t)\}\mathbb{E}\left\{\left(w(t)n_j(t)\right)^*\right\} \\
&= 0.
\end{aligned}
\tag{16}
$$

Then, Equation (14) is proven.　□

**Corollary 2.** *Considering an independent and S$\alpha$S $(N_1 + N_2) \times 1$ impulsive noise $\mathbf{n}(t)$, according to Equations (2) and (7), we have $\mathbf{y}(t) = w(t)(\mathbf{A}\mathbf{s}(t) + \mathbf{n}(t))$. Then, the pseudocovariance matrix of the proposed Co-IN strategy can be expressed as*

$$
\mathbf{R}_{Co\text{-}IN} = \mathbf{A}\boldsymbol{\Phi}_S\mathbf{A}^H + \kappa\mathbf{I}_{N_1+N_2},
\tag{17}
$$

*where $\mathbf{A}$ is the directional matrix defined in Equation (2), $\kappa$ is a constant, $\mathbf{I}_{N_1+N_2}$ is the identity matrix, and $\boldsymbol{\Phi}_S$ is the diagonal matrix of the signal vector.*

**Proof.** $\mathbf{R}_{\text{Co-IN}}$ is a $(N_1 + N_2) \times (N_1 + N_2)$ matrix; the $(i, j)$-th element is

$$
\begin{aligned}
\mathbf{R}_{\text{Co-IN}}^{(i,j)} &= \mathbb{E}\left\{y_i(t)y_j^H(t)\right\} \\
&= \mathbb{E}\left\{w(t)z_i(t)w^*(t)z_j^*(t)\right\}.
\end{aligned}
\tag{18}
$$

Substituting $z_i(t) = \sum\limits_{q=1}^{Q} e^{\frac{-j2\pi L_i \sin\theta_q}{\lambda}} s_q(t) + n_i(t)$ and $z_j(t) = \sum\limits_{p=1}^{Q} e^{\frac{-j2\pi L_j \sin\theta_p}{\lambda}} s_p(t) + n_j(t)$ into Equation (18), we then have

$$
\begin{aligned}
\mathbf{R}_{\text{Co-IN}}^{(i,j)} = \mathbb{E} & \left\{ \begin{array}{l} w^2(t) \left( \sum\limits_{q=1}^{Q} e^{\frac{-j2\pi L_i \sin\theta_q}{\lambda}} s_q(t) + n_i(t) \right) \\ \times \left( \sum\limits_{p=1}^{Q} e^{\frac{-j2\pi L_j \sin\theta_p}{\lambda}} s_p(t) + n_j(t) \right)^* \end{array} \right\} \\
= & \sum\limits_{q=1}^{Q} e^{\frac{-j2\pi L_i \sin\theta_q}{\lambda}} \times \mathbb{E} \left\{ w^2(t) s_q(t) \left( \sum\limits_{p=1}^{Q} e^{\frac{-j2\pi L_j \sin\theta_p}{\lambda}} s_p(t) + n_j(t) \right)^* \right\} \\
& + \mathbb{E} \left\{ w^2(t) n_i(t) \left( \sum\limits_{p=1}^{Q} e^{\frac{-j2\pi L_j \sin\theta_p}{\lambda}} s_p(t) + n_j(t) \right)^* \right\}.
\end{aligned}
\tag{19}
$$

Since the noise and signal are independent of each other, it has the following property: $\forall i \neq j$, $\mathbb{E}\{n_i(t)n_j(t)^*\} = 0$, $\mathbb{E}\{s_i(t)s_j(t)^*\} = 0$, and $\forall i, j$, $\mathbb{E}\{s_i(t)n_j(t)^*\} = 0$, $\mathbb{E}\{n_i(t)s_j(t)^*\} = 0$. We can obtain

$$
\begin{aligned}
\mathbf{R}_{\text{Co-IN}}^{(i,j)} &= \sum\limits_{q=1}^{Q} e^{\frac{-j2\pi L_i \sin\theta_q}{\lambda}} \mathbb{E} \left\{ w^2(t) s_q(t) \sum\limits_{p=1}^{Q} e^{\frac{j2\pi L_j \sin\theta_p}{\lambda}} s_p^*(t) \right\} \\
&\quad + \mathbb{E} \left\{ w^2(t) n_i(t) n_j^*(t) \right\} \\
&= \sum\limits_{q=1}^{Q} e^{\frac{-j2\pi L_i \sin\theta_q}{\lambda}} \mathbb{E} \left\{ w^2(t) s_q(t) s_q^*(t) \right\} e^{\frac{j2\pi L_j \sin\theta_q}{\lambda}} + \kappa \delta_{ij} \\
&= \sum\limits_{q=1}^{Q} e^{\frac{-j2\pi L_i \sin\theta_q}{\lambda}} \phi_q e^{\frac{j2\pi L_j \sin\theta_q}{\lambda}} + \kappa \delta_{ij} \\
&= \mathbf{A}_i \mathbf{\Phi}_S \mathbf{A}_j^H + \kappa \delta_{ij},
\end{aligned}
\tag{20}
$$

where $\mathbf{\Phi}_S = diag\{\phi_1, \cdots, \phi_Q\}$ is the signal covariance matrix, $\phi_q = \mathbb{E}\{w^2(t)s_q(t)s_q^*(t)\}$, and $\kappa = \mathbb{E}\{w^2(t)n_i(t)n_j^*(t)\}$; $\delta_{ij}$ is the Kronecker delta function and $\mathbf{A}_i$ is the $i$-th row of elements of $\mathbf{A}$.

Then, the pseudocovariance matrix can be written as

$$
\begin{aligned}
\mathbf{R}_{\text{Co-IN}} &= \left[ \mathbf{A}_1; \cdots; \mathbf{A}_{N_1+N_2} \right] \mathbf{\Phi}_S \left[ \mathbf{A}_1^H, \cdots, \mathbf{A}_{N_1+N_2}^H \right] + \kappa \mathbf{I}_{N_1+N_2} \\
&= \mathbf{A} \mathbf{\Phi}_S \mathbf{A}^H + \kappa \mathbf{I}_{N_1+N_2}.
\end{aligned}
\tag{21}
$$

Thus, Equation (17) holds. □

### 3.3. Algorithm Flow of Co-IN Strategy

By vectorizing Equation (17), a virtual single snapshot vector can be obtained as follows:

$$
\mathbf{r} = vec(\mathbf{R}_{\text{Co-IN}}) = \tilde{\mathbf{A}} \mathbf{b} + vec(\kappa \mathbf{I}_{N_1+N_2}),
\tag{22}
$$

where $\tilde{\mathbf{A}} = \left[ \mathbf{a}^*(\theta_1) \otimes \mathbf{a}(\theta_1), \cdots, \mathbf{a}^*(\theta_Q) \otimes \mathbf{a}(\theta_Q) \right]$, with $\otimes$ being the Kronecker product; $\mathbf{b}$ is the equivalent single snapshot signal power, whose elements are the diagonal entries of $\mathbf{\Phi}_S$. According to [34], the virtual difference coarray vector $\mathbf{r}$ contains redundant information. Therefore, by removing the redundant terms, we can obtain

$$
\tilde{\mathbf{r}} = \tilde{\mathbf{A}}_1 \tilde{\mathbf{b}} + \tilde{\kappa},
\tag{23}
$$

with the directional matrix $\tilde{\mathbf{A}}_1$, thereby corresponding to a ULA with sensor positons $[-M, M]$, where $M = N_1 N_2 + N_2 - 1$. $\tilde{\boldsymbol{\kappa}}$ is a vector obtained by removing the redundancy in $vec(\kappa \mathbf{I}_{N_1+N_2})$. Since $\tilde{\mathbf{r}}$ is a single snapshot vector, the spatial smoothing method should be applied. $\tilde{\mathbf{r}}$ can be divided into $M$ overlapping subarrays $\tilde{\mathbf{r}}_m = \tilde{\mathbf{r}}(m : m + M - 1, :)$, $m = 1, \cdots, M$. Then, a new covariance matrix with a recovered rank can be obtained by

$$\mathbf{R}_{new} = \frac{1}{M} \sum_{m=1}^{M} \tilde{\mathbf{r}}_m \tilde{\mathbf{r}}_m^H. \tag{24}$$

Consequently, by performing the eigenvalue decomposition (EVD) of $\mathbf{R}_{new}$, DOA estimation can be obtained by using the corresponding subspace-based algorithm. In this paper, we utilize the total least square ESPRIT (TLS-ESPRIT, [4]) algorithm to estimate the DOAs of the signals, and the pseudoprocedure of the proposed Co-IN method is provided in Algorithm 1.

---

**Algorithm 1:** Proposed Co-IN method.

---

**Input:** $\mathbf{z}(t) \in \mathbb{C}^{(N_1+N_2) \times 1}, t = 1, 2, \cdots, T_s$;

**for** $t = 1 : T_s$ **do**

    Calculate the local information entropy $H_t$ according to Equation (10);

    Calculate the adaptive kernel according to Equation (9);

    Calculate the memoryless normalized weights $w(t)$ according to Equation (8);

    Calculate the normalized received data: $\mathbf{y}(t) = w(t)\mathbf{z}(t)$.

**end**

1: Construct the pseudo-covariance matrix $\mathbf{R}_{\text{Co-IN}}$ and obtain the new covariance matrix $\mathbf{R}_{new}$ according to Equations (13) and (24), respectively.

2: Perform an EVD operation on $\mathbf{R}_{new}$ to obtain the signal subspace $\mathbf{E}_s = [\mathbf{e}_1, \mathbf{e}_2, \cdots, \mathbf{e}_Q]$ spanned by the eigenvectors corresponding to the first $Q$ eigenvalues.

3: Decompose $\mathbf{E}_s$ to $\mathbf{E}_x$ and $\mathbf{E}_y$, as follows

$$\mathbf{E}_s = \begin{bmatrix} \mathbf{E}_x \\ \text{The last row of } \mathbf{E}_s \end{bmatrix} = \begin{bmatrix} \text{The first row of } \mathbf{E}_s \\ \mathbf{E}_y \end{bmatrix}.$$

4: Construct a new signal matrix $\tilde{\mathbf{E}} = [\mathbf{E}_x, \mathbf{E}_y]$, and calculate $\tilde{\mathbf{R}} = \tilde{\mathbf{E}}^H \tilde{\mathbf{E}} \in \mathbb{C}^{2Q \times 2Q}$;

5: Perform EVD on $\tilde{\mathbf{R}}$ to obtain the matrix of eigenvectors $\mathbf{E}_V$, which is decomposed to

$$\mathbf{E}_V = \begin{bmatrix} \mathbf{E}_1 & \mathbf{G}_x \\ \mathbf{E}_2 & \mathbf{G}_y \end{bmatrix}, \text{ and } \mathbf{E}_1, \mathbf{E}_2, \mathbf{G}_x, \mathbf{G}_y \in \mathbb{C}^{Q \times Q}.$$

6: Calculate $\boldsymbol{\psi} = -\mathbf{G}_x \mathbf{G}_y^{-1}$;

7: Perform EVD on $\boldsymbol{\psi}$ to obtain the eigenvalue $\psi_q, q = 1, 2, \cdots, Q$;

8: Estimate the DOAs: $\theta_q = -\arcsin\left(\frac{angle(\psi_q)}{\pi}\right) \cdot \frac{180°}{\pi}, q = 1, 2, \cdots, Q$.

**Output:** The estimated DOAs $\theta_q, q = 1, 2, \cdots, Q$.

---

## 4. Simulation Results

In this section, the proposed Co-IN method is compared with some recently reported methods in the case of impulsive noise, such as the SCM [16], IN [18], ZMNL [19], GCO [21], and Toeplitz-PFLOM [36] methods. These various methods are applied to DOA estimation using the TLS-ESPRIT method. Impulsive noise is spatiotemporally uncorrelated and obeys the S$\alpha$S distribution or the Gaussian mixture model.

To evaluate the performance of the proposed algorithm and other methods, the root mean square error (RMSE) is defined as

$$\text{RMSE} = \sqrt{\frac{1}{\text{MC}} \sum_{j=1}^{\text{MC}} \frac{1}{Q} \sum_{q=1}^{Q} \left(\tilde{\theta}_{qj} - \theta_q\right)^2}, \tag{25}$$

where $\tilde{\theta}_{qj}$ denotes the DOA estimation of the $q$-th source $\theta_q$ at the $j$-th Monte Carlo (MC) experiment.

### 4.1. Complexity Analysis

In this section, the number of multiplications of the real (or complex) values is considered as a complexity criterion. The computational complexity of the proposed algorithm mainly includes the computation of the new covariance matrix $\mathbf{R}_{new}$ of $O\{(N_1 N_2 + N_2)^3\}$, the complexity of the local correlation entropy $H_t$ of $O\{2PT_s\}$, the complexity of the adaptive kernel function $\sigma_t$ of $O\{T_s\}$, the complexity of the weight function $w(t)$ of $O\{T_s\}$, the complexity of the normalized received array data $\mathbf{y}(t)$ of $O\{PT_s\}$, and the complexity of the pseudocovariance matrix $\mathbf{R}_{\text{Co-IN}}$ of $O\{P^2 T_s\}$, with a total computational complexity of $O\{(N_1 N_2 + N_2)^3 + (P^2 + 3P + 2)T_s\}$. For the proposed Co-IN algorithm with ULA, it has a total computational complexity of $O\{(P^2 + 3P + 2)T_s\}$. The main computational complexity results of the proposed algorithm and other algorithms are given in Table 1. It can be seen from Table 1 that the complexity of the proposed algorithm with ULA is slightly higher than that of the IN, ZMNL, and SCM algorithms, but it is significantly lower than those of the GCO and Toeplitz-FLOM methods. Moreover, the DOA estimation performance of the proposed method can be verified in the later simulations.

**Table 1.** Complexity analysis of the covariance matrix before using TLS-ESPRIT.

| Algorithm | Total Computational Complexity | Remarks |
|:---:|:---:|:---:|
| IN with ULA [18] | $O\{(P^2 + P + 1)T_s\}$ | $P = N_1 + N_2$ |
| GCO with ULA [21] | $O\{(6P^2 + 2P + 2)T_s\}$ | $P = N_1 + N_2$ |
| ZMNL with ULA [19] | $O\{(P^2 + 2P + 2)T_s\}$ | $P = N_1 + N_2$ |
| SCM with ULA [16] | $O\{(P^2 + 2P)T_s\}$ | $P = N_1 + N_2$ |
| Toeplitz-FLOM with ECA [36] | $O\{(MN + M)^3 + 3P^2 T_s\}$ | $P = 2M + N - 1$ |
| Proposed Co-IN with ULA | $O\{(P^2 + 3P + 2)T_s\}$ | $P = N_1 + N_2$ |
| Proposed Co-IN with NA | $O\{(N_1 N_2 + N_1)^3 + (P^2 + 3P + 2)T_s\}$ | $P = N_1 + N_2$ |

### 4.2. SαS Impulsive Noise

The generalized signal-to-noise ratio (GSNR) of an impulsive noise with an *SαS* distribution [12] is defined as

$$\text{GSNR} = 10 \log \left( \frac{\mathbb{E}\{|\mathbf{s}(t)|^2\}}{\gamma} \right), \tag{26}$$

where $\gamma$ is the dispersion parameter.

Consider two independent quadrature phase shift keying (QPSK) sources with DOAs of $10° \times rand(1,2) + [-20°, 15°]$. Other parameters are set as follows: the total number of ULA is 10 (SCM, IN, GCO, ZMNL and Co-IN-ULA), the extended coprime array is $2M = 6$ with $N = 5$ (Toeplitz-PFLOM), and the nested array is $N_1 = N_2 = 5$ (the proposed Co-IN with NA). In addition, $p = 0.9$ is selected in the GCO method. In addition, to ensure a fair comparison of the DOA estimation performance of different algorithms, we extended all the above ULA-based algorithms to NA scenarios (where the total number of physical sensors is always identical), which are denoted as SCM-NA, IN-NA, GCO-NA, ZMNL-NA, and GCO-NA. The difference between these algorithms and the proposed algorithm lies in the construction of the equivalent covariance matrix Equation (18), and all of them can realize DOA estimation using the nested array technique.

Firstly, we considered the scenario where $\alpha$ is between 0.3 and 1.4, set GSNR = 4 dB, set the snapshots to $T_s = 400$, and set MC = 1000; the simulation results of all the compared algorithms are shown in Figure 3. As can be seen from Figure 3, the RMSE of all the algorithms gradually decreased as $\alpha$ increased. Moreover, the RMSE of the proposed

algorithms was smaller than that of the other compared methods for either ULA or NA scenarios. From Figure 3b, it can be found that the GCO algorithm was more adaptable to NA scenes compared to the ZMNL algorithm. Under the same intensity of impulsive noise, the proposed algorithm provided better performance than the SCM, GCO, IN, ZMNL and Toeplitz-PFLOM algorithms.

Then, we tested the RMSE performance of all the compared algorithms versus their GSNRs and snapshots, where $\alpha = 0.8$. As shown in Figure 4a, the proposed Co-IN algorithm and IN-ULA algorithm had better estimation performance than other methods under the same GSNR. Similar performance results can be observed in Figure 4c, where the number of snapshots was variable. In addition, we extended the aforementioned comparison algorithms to NA scenarios and compared the performance with the proposed Co-IN-NA algorithm among different GSNRs and numbers of snapshots, as shown in Figure 4b,d. As can be seen from Figure 4b,d, the RMSE of all the methods was significantly reduced as compared to Figure 4a,c. Also, it can be observed that the SCM-NA, IN-NA, and the proposed Co-IN-NA algorithms outperformed the other algorithms, where the proposed algorithm has the best performance result.

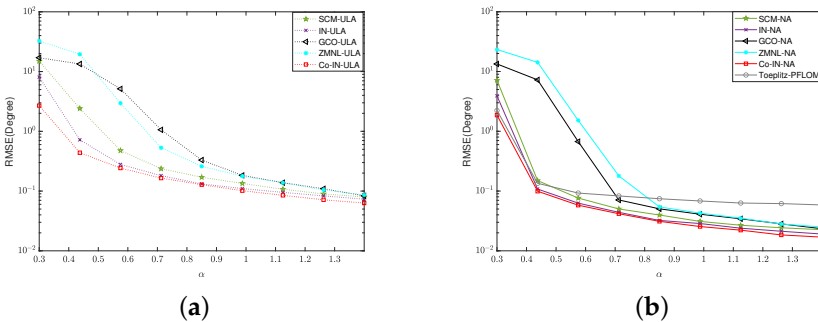

**Figure 3.** RMSE of DOA estimation versus $\alpha$, where GSNR = 4 dB, $T_s = 400$, MC = 1000. (**a**) ULA. (**b**) Nested arrays.

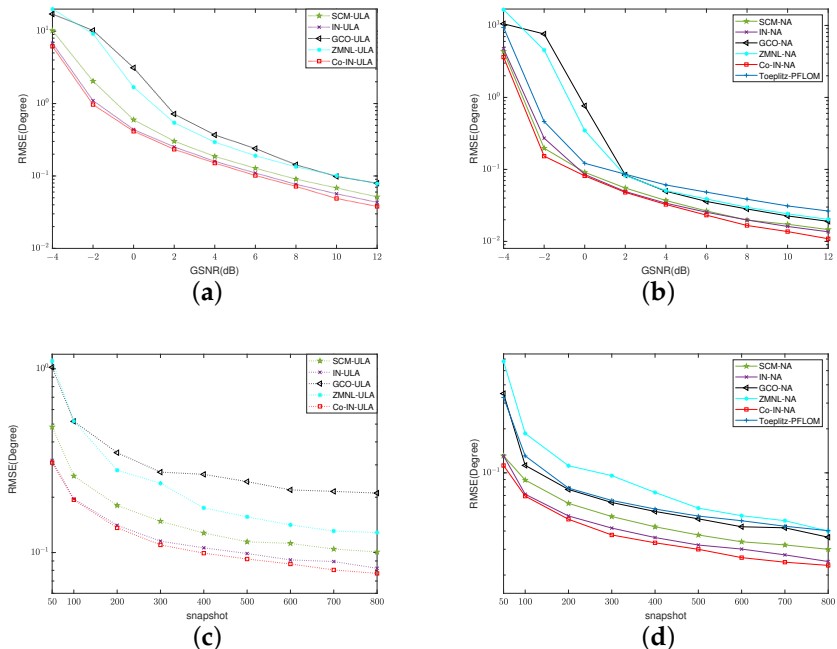

**Figure 4.** RMSE of DOA estimation versus (**a**) GSNRs with ULA, where $T_s = 400$, MC = 1000. (**b**) GSNRs with NA, where $T_s = 400$, MC = 1000, (**c**) Snapshots with ULA, where GSNR = 4 dB, MC = 1000. (**d**) Snapshots with NA, where GSNR = 4 dB, MC = 1000.

For the IN algorithm, the proposed Co-IN-ULA method shown in Figure 4a yielded a slightly better performance with $\alpha = 0.8$. To better represent the Co-IN algorithm's effectiveness, Figure 5 compares the two algorithms with impulsive noise in the $\alpha$ range of 0.2 to 1.4; the GSNR was taken between $-2$ dB and 6 dB, and the other experimental conditions were the same as in Figure 4. As Figure 5 shows, in the case of the ULA, the Co-IN-ULA algorithm's estimation accuracy was significantly higher than the IN algorithm for the same values of $\alpha$. Moreover, the proposed Co-IN algorithm was applicable to the strongly impulsive noise environment.

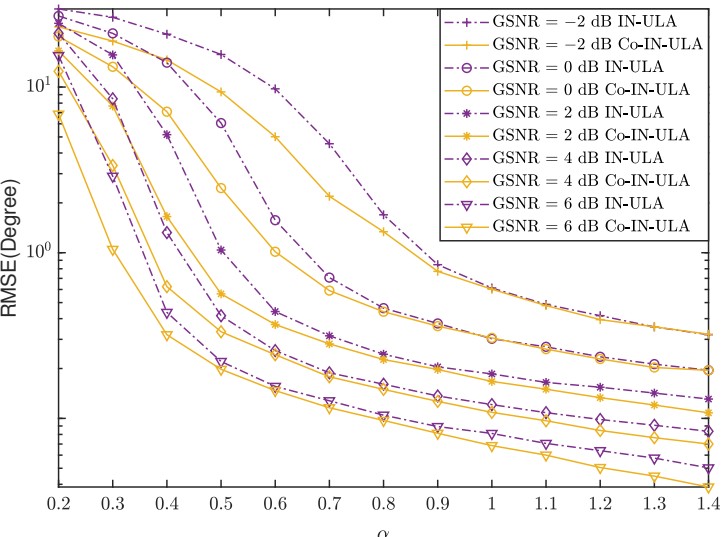

**Figure 5.** RMSE of DOA estimation versus $\alpha$, $T_s = 400$, MC = 1000.

### 4.3. Gaussian Mixture Heavy-Tailed Noise

We adopted the two-term GMM [24] with the following probability density function:

$$p_n(x) = \sum_{h=1}^{2} \frac{c_h}{\pi \sigma_h^2} \exp\left(-\frac{|x|^2}{\sigma_h^2}\right) \tag{27}$$

where $0 \leqslant c_h \leqslant 1$ and $\sigma_h^2$ denote the probability and variance of the $h$-th Gaussian term, respectively. We set $\sigma_2^2 = 100\sigma_1^2$, $c_1 > c_2 > 0$, and $c_1 + c_2 = 1$. According to [24], the SNR can be defined as

$$\text{SNR} = \frac{\mathbb{E}\left\{|\mathbf{s}(t)|^2\right\}}{\sigma_1^2}. \tag{28}$$

Figure 6a illustrates the RMSE performance results of the proposed algorithm compared with other algorithms versus the SNRs in the GMM environment, and Figure 6c shows the RMSE results for different numbers of snapshots. The GMM parameters were $c_1 = 0.9$, $c_2 = 0.1$ and $\sigma_1^2 = 1$, $\sigma_2^2 = 100$, and the other parameters were consistent with the simulation in Section 4.2. It can be seen from Figure 6a that the proposed Co-IN algorithm had the best RMSE performance over its three competitors under noise following the same SNR in the presence of the Gaussian mixture model. The RMSE performance versus the number of snapshots shown in Figure 6c also verifies the superiority of the proposed methods. Furthermore, Figure 6b,d show the RMSE results of the methods in Figure 6a,c in the presence of nested arrays. It can be seen that in the context of nested arrays, the performance of all the algorithms improved with the increase in the SNR or the number of snapshots. From the figures, it can be seen that the proposed Co-IN-NA algorithm outperformed the other algorithms, which also illustrates the effective resistance of the Co-IN strategy against impulsive noise.

Figure 6a shows that the RMSE of the Co-IN-ULA algorithm was similar to the IN algorithm for the values of $c_2 = 0.1$. To clarify the advantage of the proposed Co-IN algorithm, we selected GMM noise with different parameters of $c_2$ for comparison, as shown in Figure 7, where $c_2$ ranged from 0.04 to 0.36, the number of snapshots was $T_s = 400$, MC = 1000, the sensor numbers of the ULA were 10, the SNR was taken between −8dB and 0dB, and the other experimental conditions were the same as in Figure 6a.

The results in Figure 7 indicate that the proposed Co-IN algorithm can effectively handle different components of Gaussian mixture heavy-tailed noise. In addition, the Co-IN-ULA algorithm is a simple application of our proposed algorithm to the ULA. The focus of our work is to propose the Co-IN-NA algorithm for nested array configurations.

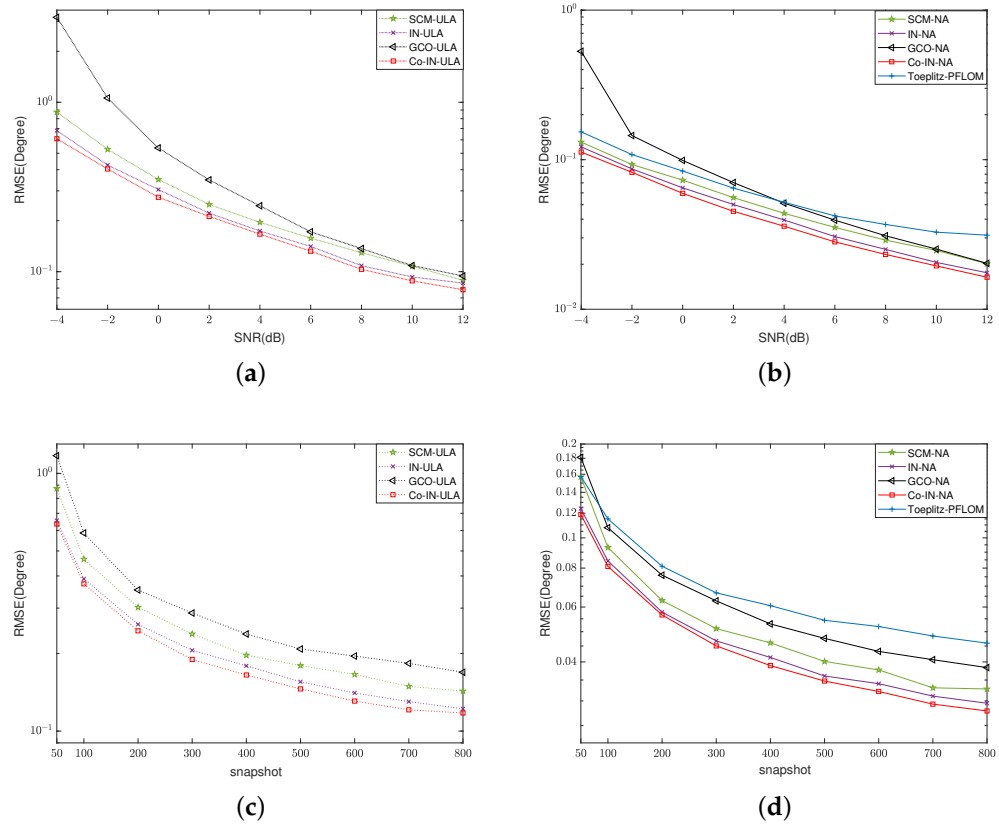

**Figure 6.** RMSE of DOA estimation versus (**a**) SNRs with ULA, where $T_s = 400$, MC = 1000. (**b**) SNRs with NA, where $T_s = 400$, MC = 1000. (**c**) Snapshots with ULA, where SNR = 4 dB, MC = 1000. (**d**) Snapshots with NA, where SNR = 4 dB, MC = 1000.

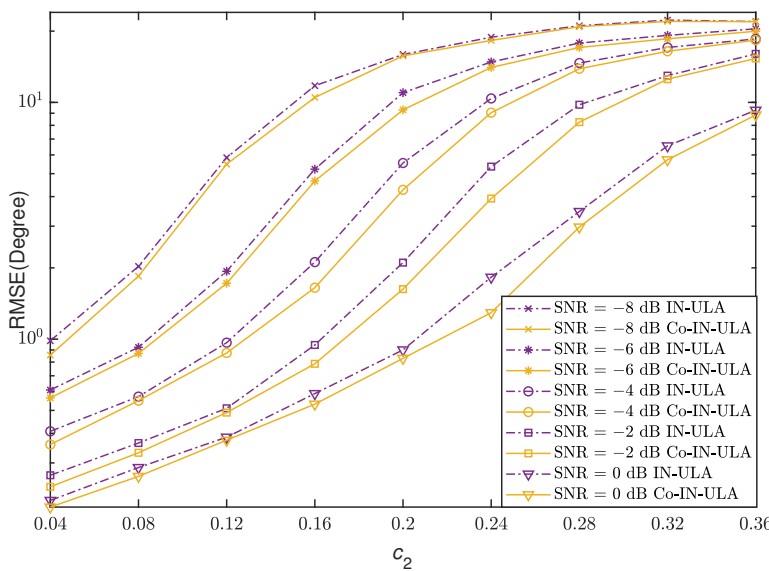

**Figure 7.** RMSE of DOA estimation versus $c_2$, $T_s = 400$, MC = 1000.

### 4.4. Underdetermined DOA Estimation

The ULA-based method is no longer applicable when the number of sources exceeds the number of sensors, and only the Toeplitz-PFLOM (with degrees of freedom (DOFs) $M(N + 1) - 1$) method could be considered in comparison with the proposed Co-IN-NA (with DOFs $N_1(N_2 + 1) - 1$) method. In this simulation, the MUSIC pseudospectrums of the proposed algorithm and Toeplitz-PFLOM were tested by different DOA values and $\alpha$ values. The number of snapshots was 1000, and the GSNR = 20 dB; these settings were defined in the following three cases:

Case 1: $\theta$ is taken between $[-48°, 48°]$ with an interval of 8°, $Q = 13$, and $\alpha = 0.5, 1.2$.
Case 2: $\theta$ is taken between $[-48°, 48°]$, with an interval of 6°, $Q = 17$, and $\alpha = 0.5, 1.2$.
Case 3: $\theta$ is taken between $[-55°, 55°]$, with an interval of 5°, $Q = 23$, and $\alpha = 0.5, 1.2$.

Figures 8 and 9 show the MUSIC pseudospectrums of the proposed Co-IN-NA and Toeplitz-PFLOM methods for different source numbers and characteristic exponents of $\alpha$. Figure 8 illustrates the performance of the Co-IN-NA and Toeplitz-PFLOM methods for the number of sources ($Q = 13$) being slightly larger than the number of sensors ($2M + N - 1 = N_1 + N_2 = 10$). The Co-IN-NA shown in Figure 8 provided an accurate estimate, while the Toeplitz-PFLOM estimation had an incorrect peak at a small value of $\alpha$. Figure 9 shows the MUSIC spectrum with the number of sources ($Q = 17$) reaching the upper limit estimated by the Toeplitz-PFLOM method (DOFs = 17). It can be seen that the Toeplitz-PFLOM method showed a failure of the true DOA for different $\alpha$ values. By comparison, the proposed Co-IN-NA method had improved estimation results. Compared to Figure 9, Figure 10 shows the MUSIC spectrum for 23 sources; at this time, the Toeplitz-PFLOM method was no longer working, while the Co-IN-NA method was still robust in all cases.

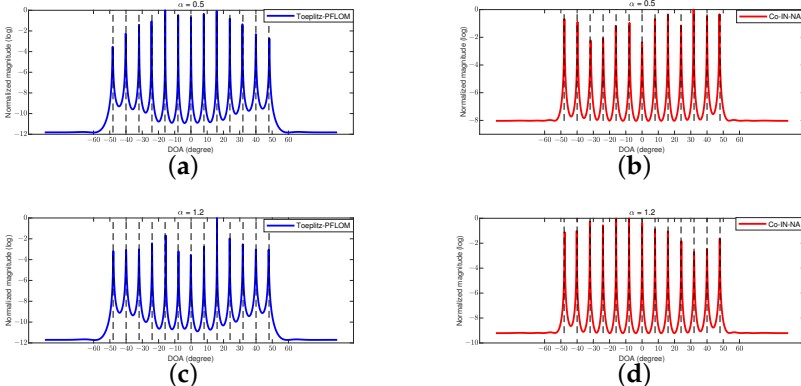

**Figure 8.** Case 1: (**a**) Toeplitz-PFLOM method with $\alpha = 0.5$; (**b**) Co-IN-NA method with $\alpha = 0.5$; (**c**) Toeplitz-PFLOM method with $\alpha = 1.2$; (**d**) Co-IN-NA method with $\alpha = 1.2$. Where the dashed line denotes the true DOAs.

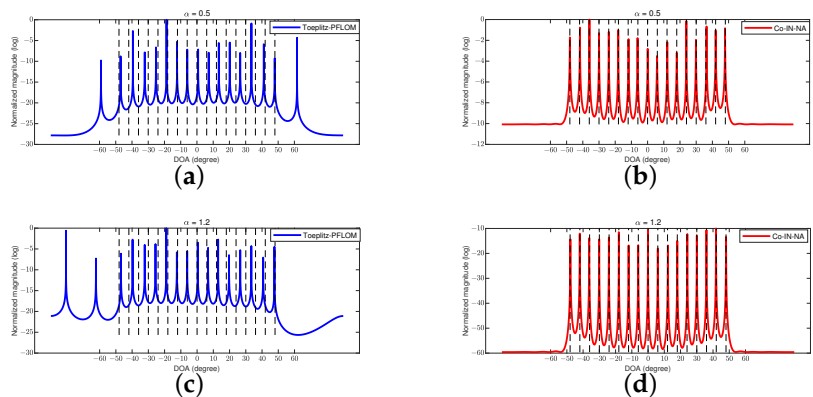

**Figure 9.** Case 2: (**a**) Toeplitz-PFLOM method with $\alpha = 0.5$; (**b**) Co-IN-NA method with $\alpha = 0.5$; (**c**) Toeplitz-PFLOM method with $\alpha = 1.2$; (**d**) Co-IN-NA method with $\alpha = 1.2$. Where the dashed line denotes the true DOAs.

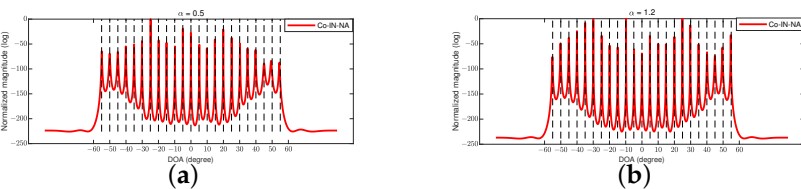

**Figure 10.** Case 3: (**a**) $\alpha = 0.5$; (**b**) $\alpha = 1.2$. Where the dashed line denotes the true DOAs.

In addition, we tested the performance of the Co-IN-NA algorithm for the different number of sources. Figure 11 depicts the performance dependence of the proposed Co-IN-NA algorithm on the number of sources, where the total number of array sensors was set to 10, i.e., $N_1 = N_2 = 5$. In addition, the characteristic exponents were $\alpha = 0.6$ and $\alpha = 0.9$, the number of snapshots was $T_s = 400$, and MC = 1000. Thanks to the advantage of the nested array technique, it can be seen from Figure 11 that our method still obtained satisfactory performance when the number of sources was $Q = 12, 15$. The RMSE was within one degree when the GSNR > 2 dB. This simulation indicates the applicability of the Co-IN-NA algorithm to the variation in source number. Further work will consider the extension of the proposed algorithm to other sparse array scenarios.

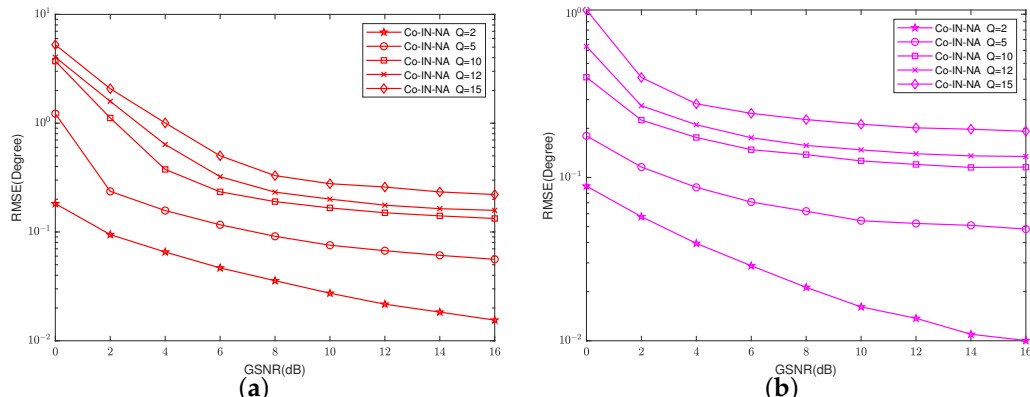

**Figure 11.** RMSE of DOA estimation versus different numbers of sources Q; $T_s = 400$; MC = 1000. (**a**) $\alpha = 0.6$; (**b**) $\alpha = 0.9$.

## 5. Conclusions

In this paper, a correlation entropy-based infinite norm preprocessing algorithm was proposed for DOA estimation with nested arrays in the presence of impulsive noise. The proposed method utilizes an exponential kernel function for each snapshot of the sensor received array data, which effectively suppresses the impulsive noise outliers and constructs an equivalent covariance matrix. Not only doe this improve the accuracy and robustness of the DOA estimation under impulsive noise, but it also can detect more sources than sensors. Meanwhile, the proposed method does not require any prior knowledge of impulsive noise information (e.g., the characteristic exponent $\alpha$). Simulation results through two different noise scenarios show that the proposed method outperformed the conventional IN, ZMNL and SCM approaches and the recently reported GCO and Toeplitz-PFLOM methods in DOA estimation. However, the work in this paper does not consider the mutual coupling effect between the dense array elements, and it will be our future work to find out how to efficiently decouple these elements.

**Author Contributions:** Conceptualization, X.D. and J.Z.; Methodology, X.D. and J.Z.; writing—original draft preparation, J.Z.; Writing—review and editing, M.S. and Y.W.; Supervision, Y.W. and R.G.; Funding acquisition, M.S. All authors have read and agreed to the published version of the manuscript.

**Funding:** This work is supported by the National Natural Science Foundation of China under Grant 62101250, Grant 62101251, and the Natural Science Foundation of Jiangsu Province under Grant BK20210281 and China Postdoctoral Science Foundation No.2023M732965.

**Data Availability Statement:** The data used in this paper can be requested from the corresponding authors upon request.

**Conflicts of Interest:** Author Sun Meng is a postdoctoral student at Nanjing Panda Electronics Company Limited. He is also a professor at Nanjing University of Aeronautics and Astronautics. The remaining authors declared that the research was conducted in the absence of any commercialor financial relationships that could be construed as a potential conflict of interest. The funder had no role in the design of the study, in the collection, analyses, or interpretation of data, in the writing of the manuscript, or in the decision to publish the results.

## Appendix A

**Proof of Equation (12).** Let $\mathbf{R}_{\text{Co-IN}}^{(i,j)}$ denote the $(i, j)$-th element of matrix $\mathbf{R}_{\text{Co-IN}}$; since $\mathbf{R}_{\text{Co-IN}}^{(i,j)}$ is a complex number, we prove separately that the real part $\mathcal{R}\left\{\mathbf{R}_{\text{Co-IN}}^{(i,j)}\right\}$ and imaginary part $\mathcal{I}\left\{\mathbf{R}_{\text{Co-IN}}^{(i,j)}\right\}$ are bounded. According to Equations (18)–(20), we have

$$\mathcal{R}\left\{\mathbf{R}_{\text{Co-IN}}^{(i,j)}\right\} = \mathcal{R}\left\{\mathbb{E}\left\{y_i(t)y_j^*(t)\right\}\right\}$$
$$= \mathcal{R}\left\{\mathbf{A}_i \mathbf{\Phi}_S \mathbf{A}_j^H + \kappa\delta_{ij}\right\} \tag{A1}$$

For any complex number $c$, $\mathcal{R}\{c\} \leqslant |c|$, we then have

$$\mathcal{R}\left\{\mathbf{A}_i \mathbf{\Phi}_S \mathbf{A}_j^H + \kappa\delta_{ij}\right\}$$
$$\leqslant \left|\mathbf{A}_i \mathbf{\Phi}_S \mathbf{A}_j^H + \kappa\delta_{ij}\right| \tag{A2}$$
$$\leqslant \left|\mathbf{A}_i \mathbf{\Phi}_S \mathbf{A}_j^H\right| + |\kappa|$$

Under the assumption that the signal distribution has finite covariance, then $\mathbf{\Phi}_S$ is bounded, i.e.,

$$\left|\mathbf{A}_i \mathbf{\Phi}_S \mathbf{A}_j^H\right| < \infty, \forall i, j. \tag{A3}$$

Hence, the boundness of $\mathcal{R}\left\{\mathbf{R}_{\text{Co-IN}}^{(i,j)}\right\}$ is limited to the boundness of $|\kappa|$. According to *Corollary 1*, we have $|\kappa| = \left|\mathbb{E}\left\{w^2(t)n_i(t)n_j^*(t)\right\}\right| = |D_{i,j}| < \infty$, where $D_{i,j}$ is the $(i, j)$-th element of matrix $\mathbf{D}$.

Thus, $\mathcal{R}\left\{\mathbf{R}_{\text{Co-IN}}^{(i,j)}\right\} < \infty$ holds, i.e., $\mathcal{R}\left\{\mathbf{R}_{\text{Co-IN}}^{(i,j)}\right\}$ is bounded. The proof of the imaginary part $\mathcal{I}\left\{\mathbf{R}_{\text{Co-IN}}^{(i,j)}\right\}$ is the same as the real part. $\square$

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
