# Peer review of "Direction of Arrival Estimation with Nested Arrays in Presence of Impulsive Noise: A Correlation Entropy-Based Infinite Norm Strategy"

_remotesensing, doi:10.3390/rs15225345_

Round 1
Reviewer 1 Report
Comments and Suggestions for Authors
This manuscript proposes a correlation entropy-based infinite norm preprocessing algorithm, which is applicable to the impulsive noise. The detailed comments are shown as follows.
1. The authors claim that the proposed method is data-driven and zero-memory as the contribution. However, the proposed method is apparently model-based, and the authors did not provide any explaination about why the proposed method is zero-memory.
2. The authors claim that the impulsive noise should obey a spherically symmetric distribution on the complex plane. The reviewer wonders whether this is reasonable in practical scenarios, please clarify.
3. The literature review can be updated, and some most recent advances in the field of sparse arrays can be reviewed or discused in the Introduction, such as
- Structured Nyquist Correlation Reconstruction for DOA Estimation with Sparse Arrays, IEEE Transactions on Signal Processing, 2023.
- Coarray Tensor Direction-of-Arrival Estimation, IEEE Transactions on Signal Processing, 2023.
- Structured Tensor Reconstruction for Coherent DOA Estimation, IEEE Signal Processing Letters, 2022.
4. To ensure a fair comparison of the DOA estimation performance of different algorithms, it is necessary for all algorithms to employ the same array structure. The authors have compared the DOA estimation performance of different algorithms using ULA in Fig 3, 4, 6, but omitted the performance comparisons using nested arrays.
Comments on the Quality of English LanguageN/A
Reviewer 2 Report
Comments and Suggestions for Authors
In this paper, a correlation entropy-based infinite norm preprocessing algorithm is proposed for DOA estimation with nested arrays in the presence of impulsive noise. The work is interesting, however, some problems need to be addressed.
1. DOA estimation algorithms are not well focussed, explained, and referred to (The references [1]-[5] are about traditional DOA estimation). There are subspace methods and sparsity-based methods in the literature. Some sparsity-based (but not limited to) references that enhance the literature survey of the paper are::
-…,“ Iterative implementation method for robust target localization in a mixed interference environment”, 2021.
-…,“An iterative dictionary learning-based algorithm for DOA estimation”, 2016.
-…,“An Iterative Implementation-Based Approach for Joint Source Localization and Association Under Multipath Propagation Environments”, 2023.
…,“Target Height Measurement under Complex Multipath Interferences without Exact Knowledge on the Propagation Environment”, 2022.
2. Improved performance usually comes at a cost of some kind. This question needs to be addressed.
3. Please explain if the proposed algorithm can be applied to the sparse linear array situation, such as coprime arrays.
4. There are some format errors and typos throughout the paper, such as line 65.
Comments on the Quality of English LanguageIt would be great if the authors could improve the English expression and organization, as there are some format and expression errors.
Reviewer 3 Report
Comments and Suggestions for Authors
In this paper, the authors propose a correlation entropy-based infinite norm preprocessing algorithm, which is applicable to any type of impulsive noise. Simulation results demonstrate the effectiveness of the proposed method. My main concerns are listed as follows.
(1) This paper researches the DOA estimation with nested arrays in the presence of impulsive noise. In the Introduction part, the authors should add more recent references on the nested array DOA estimation.
(2) In the third paragraph of the Introduction part, it seems that the authors do not introduce relevant references in detail.
(3) In the last paragraph, the authors should only introduce their work and their main contributions.
(4) As far as I know, there are many variations of the nested arrays that can greatly increase the number of DOFs as well as reduce the mutual coupling effects. Why do the authors choose the nested array structure of Figure 1.
(5) It is well known that the nested arrays may have mutual coupling effects in some elements. How do the authors consider this effect?
(6) In the equation (24), does the effective DOFs for DOA estimation be reduced? If so, how many DOFs are reduced?
(7) In the Section of Simulation results, why does the proposed method not be compared with the nested array DOA estimation methods? I suggest the authors should add the performance comparison of the proposed method with the nested array DOA estimation methods.
Comments on the Quality of English LanguageMinor editing of English language required.
Round 2
Reviewer 3 Report
Comments and Suggestions for Authors
The authors addressed the comments well.